# The Influence of Pericardial Fat on Left Ventricular Diastolic Function

**DOI:** 10.3390/diagnostics14070702

**Published:** 2024-03-27

**Authors:** Patrícia Coelho, Hugo Duarte, Carlos Alcafache, Francisco Rodrigues

**Affiliations:** 1Higher School of Health Dr Lopes Dias-Polytechnic Institute of Castelo Branco/Escola Superior de Saúde Dr. Lopes Dias, Instituto Politécnico de Castelo Branco, 6001-909 Castelo Branco, Portugal; patriciacoelho@ipcb.pt; 2SPRINT-Sport Physical Activity and Health Research & Innovation Center/Centro de Investigação e Inovação em Desporto Atividade Física e Saúde, 6001-909 Castelo Branco, Portugal; 3Polytechnic Institute of Castelo Branco, 6001-909 Castelo Branco, Portugal; hugo-duarte13@hotmail.com (H.D.); alcafache@ipcb.pt (C.A.); 4Guarda Local Health Unit, 6270-498 Seia, Portugal

**Keywords:** pericardium, echocardiography, ventricular dysfunction, left, heart failure, diastolic, abdominal circumference

## Abstract

Background: Heart failure is a major cause of morbidity and mortality worldwide; left ventricular diastolic dysfunction plays a leading role in this clinical context. Diastolic dysfunction may be predisposed by increased abdominal fat and, consequently, increased pericardial and epicardial adiposity. This study aimed to determine whether pericardial fat (PF) and epicardial fat (EF) are associated with left ventricular diastolic function. Methods: A total of 82 patients had their abdominal circumference measured and underwent transthoracic echocardiography to measure the thickness of PF and EF and assess the left ventricular diastolic function. Two groups were created based on mean pericardial fat (PF) thickness (4.644 mm) and were related to abdominal circumference and echocardiographic parameters. Results: Subjects in the PF High group showed a significant decrease in septal e’ (*p* < 0.0001), lateral e’ (*p* < 0.0001), and E/A ratio (*p* = 0.003), as well as a significant increase in E/e’ ratio (*p* < 0.0001), E wave deceleration time (*p* = 0.013), left atrial volume (*p* < 0.0001), the left ventricle mass (*p* = 0.003), tricuspid regurgitant jet velocity (*p* < 0.0001), and the left ventricle diameter (*p* = 0.014) compared to the PF Low group. Correlations were found between pericardial fat and nine echocardiographic parameters in the study, while epicardial fat (EP) only correlated with eight. Conclusions: Measurement of abdominal circumference, PF, and EF is an early indicator of diastolic changes with transthoracic echocardiography being the gold standard exam.

## 1. Introduction

Heart failure (HF) is a chronic disease that results from the inability of the heart to perform its functions properly. It is a major cause of morbidity and mortality worldwide and left ventricular (LV) diastolic dysfunction plays a leading role in this clinical context [1,2,3,4].

The importance of early assessment of LV diastolic function in the prevention of adverse outcomes has been discussed in several studies [1,2,3,4]. The realization that approximately half of individuals with signs and symptoms of heart failure do not have systolic dysfunction, but have diastolic dysfunction has highlighted the need for the assertive and timely diagnosis of this clinical condition [5,6]. LV diastolic dysfunction is usually defined as abnormal left ventricular relaxation, which can be assessed and quantified using echocardiographic measurements and algorithms included in the American Society of Echocardiography/European Association of Cardiovascular Imaging (ASE/EACVI) guidelines [7,8].

Obesity is also a major cause of morbidity and mortality worldwide and is an independent risk factor for the development of heart disease. The increase and abnormal distribution of visceral fat in the myocardium and arteries has a strong correlation with cardiovascular disease [9,10]. Pericardial fat (PF) accumulates in the pericardium. Epicardial fat (EF) accumulates between the visceral pericardium and the myocardium. When PF is excessive, it compresses the myocardium, resulting in reduced distensibility and cardiac remodeling, which can lead to diastolic dysfunction [10]. Individuals with a high body mass index and obesity are therefore more likely to develop diastolic dysfunction [11,12]. Given the pathophysiology of these conditions, there may be an association between abdominal circumference, PF, EF, and LV diastolic dysfunction. Several studies have been carried out to assess this relationship [1,2,13,14].

Currently, the most effective way to diagnose diastolic dysfunction is transthoracic echocardiography using two-dimensional (2D) and Doppler echocardiography. To evaluate diastolic function, the ASE/EACVI [8] developed two diagnostic algorithms based on echocardiographic parameters [8].

A recent study by De Wit-Verheggen [1] used MRI to assess PF in healthy individuals. Participants were divided into two groups (based on PF levels). These groups were compared based on diastolic function, assessed using transthoracic echocardiography. The study discovered changes in some parameters, such as E/e’, E/A, septal e’, and lateral e’, which may be associated with LV relaxation and filling. However, the study also found a surprising decrease in left atrial volume (LAV) rather than the expected increase in the presence of diastolic dysfunction. The authors propose that it may be due to a limitation in left atrial (LA) enlargement caused by excessive PF. They concluded that the presence of PF alters some echocardiographic parameters of diastolic dysfunction, but more research is needed to fully understand this condition [1].

Ma W et al. (2021) [2] conducted a study to evaluate EF thickness and LV diastolic function in healthy subjects with transthoracic echocardiography. The study included two fat measurements, as follows: one performed at the length of the aortic root line, perpendicular to the right ventricular free wall (EFT1) and another performed at the maximum fat thickness perpendicular to the right ventricular free wall (EFT2), both measured in end-diastole. The results showed that patients with higher EFT1 had an increase in E/e’ and E/A, as well as a decrease in septal e’ and lateral e’, indicating a correlation between PF and diastolic dysfunction. Furthermore, an independent negative correlation was observed between patients with higher EFT2 and a decrease in mean e’, suggesting a greater correlation between maximal thickness measured perpendicular to the right ventricular free wall and diastolic dysfunction. The authors concluded that more attention should be paid to this type of fat [2].

Despite being a very interesting topic, there are few studies to determine which echocardiographic parameters related to diastolic dysfunction change in the presence of increased PF and EF. The studies carried out showed a significant variation in these results, with no uniformity or guiding line [15]. Therefore, this study becomes more important to standardize the findings and contribute to this solution.

This study aims to establish a correlation between abdominal circumference and both PF and EF, as well as between the latter and LV diastolic performance so that, in the future, it will be possible to predict the development of diastolic dysfunction and consequent diastolic heart failure based on the measurement of abdominal circumference and PF and EF. These measurements are often underestimated and ignored in transthoracic echocardiography. In this study, we used routine transthoracic echocardiography to measure all echocardiographic parameters associated with diastolic function and then related them with the thickness of PF and EF, as well as relating the latter with the abdominal circumference [1,2,3,4].

## 2. Materials and Methods

### 2.1. Study Design

This study is cross-sectional, observational, prospective, and quantitative, with a non-probabilistic sample by rational choice, including, therefore, all individuals without known pathology who underwent a convenience echocardiographic study in the echocardiography laboratory of the cardiology service of a local health service unit in the central region of Portugal between 1 August and 31 December 2022. All individuals with systolic dysfunction, cardiomyopathies, significant valvular pathology, or those with intracardiac devices were excluded from the study. Considering the above criteria, a sample of 82 individuals was collected. 

### 2.2. Protocol

The data collection procedure was carried out in two distinct phases, as follows: the first phase involved the measurement of abdominal circumference with a tape measure and the second phase involved transthoracic echocardiography using a Toshiba^®^ model Xario (Toshiba Medical Systems Corporation (Canon Medical Systems Corporation) Tochigi, Japan) device with a 2.4–4.5 MHz frequency transducer.

The abdominal circumference was measured around the abdomen just above the navel. To standardize the method, obtain more accurate measurements, and reduce bias, this measurement was always performed by the same professional [16,17,18].

A transthoracic echocardiogram was then performed according to the latest ASE/EACVI guidelines to measure PF and assess its functionality [19]. PF and EF were measured through the parasternal long-axis window in the region of the free wall of the right ventricle, both in end-systole and using the average of three consecutive beats [20]. 

The remaining echocardiographic measurements were indexed to the individual’s surface area, so the left ventricular ejection fraction (LVEF) was determined using the biplanar Simpson method and both the end-diastolic diameter and LV mass were determined using the M mode. The E wave, the E wave deceleration time (DT), and the E/A ratio were measured using pulsed Doppler with the cursor perpendicular to the mitral annulus and the sample positioned at the end of the mitral leaflets. LAV was quantified using the biplanar method. To measure septal e’ and lateral e’, two techniques were used simultaneously, namely pulsed Doppler and tissue Doppler, with the cursor positioned at the intersection of the interventricular septum and mitral annulus and the intersection of the LV lateral wall and mitral annulus, respectively. In this way, it was also possible to obtain the E/e’ ratio. Continuous Doppler was used to measure tricuspid regurgitant jet velocity (TRJV) by aligning the cursor with the tricuspid regurgitant jet and recording its maximum velocity. The diagnosis of diastolic dysfunction was made using the ASE/EACVI guidelines’ algorithm for individuals with preserved ejection fraction. To confirm this diagnosis, at least three of the four echocardiographic parameters needed to be positive, LAV, E/e’ ratio, lateral e’ or septal e’, and TRJV [8].

After collecting all variables, the sample was divided into two groups based on the average PF thickness, with the aim of facilitating data analysis and subsequent interpretation. The limit used was 4.644 mm, which represents the real value of the sample, making it more distributable. The minimum thickness observed was 1.4 mm, while the maximum was 11.6 mm. A flowchart of inclusion and exclusion criteria is provided for better understanding. (Figure 1).

### 2.3. Study Variables

To achieve the study’s objective, we gathered various quantitative echocardiographic variables, including the diameter of the PF and EF, LAV, LVEF, E/A ratio, E/e’ ratio, E-wave velocity, TRJV, DT, septal e’, lateral e’, LV mass, and LV end-diastolic diameter. Sample elements, such as sex, race, age, weight, height, BMI, and waist circumference were also collected to help characterize the data.

### 2.4. Statistical Analysis of the Sample

To assess the normality of the sample distribution, we used the Kolmogorov–Smirnov normality test. We conducted the *t*-test to compare the two groups of PF with quantitative echocardiographic variables, including E wave, DT, septal e’, lateral e’, E/e’ ratio, LAV, ejection fraction, mass of LV, and LV diameter. Additionally, we used the Mann–Whitney test to compare the E/A ratio and TRJV. The Spearman correlation test was used to investigate the correlations between abdominal circumference and PF and EF, as well as between echocardiographic parameters of diastolic function and both fats under study. To evaluate the independent association between PF and echocardiographic parameters of diastolic dysfunction, a multivariable linear regression analysis was performed on septal e’, LAV, VRJ, LVEF, LV diameter, and LV mass. These models were adjusted for sex, age, and BMI.

To better understand the relationship between the prevalence of diastolic function diagnosis and the two groups of PF, we performed a Chi-square test.

For the statistical treatment of qualitative variables, a descriptive analysis was carried out using relative (%) and absolute (*n*) frequencies, as well as measures of central tendency (mean) and dispersion (standard deviation).

The data obtained from the sample were analyzed and processed using IBM SPSS Statistics^®^ (Statistical Package for the Social Sciences) version 27 (Armonk, NY, USA), a statistical analysis software. A *p*-value of 0.05 or less was considered statistically significant for all tests.

### 2.5. Ethical Considerations

The work received a positive opinion from the Ethics Committee (98/2022). This study has strictly adhered to ethical principles and ensured the confidentiality of all data, results, and interpretations. Personal data were collected only when necessary for the study and all information gathered was treated as confidential. The data collected were solely used for academic purposes within the context of the research.

The research team declares that it has no conflicts of interest and is committed to respecting the principles expressed in the Declaration of Helsinki. This research does not have any profit or commercial purposes.

## 3. Results

The study sample consists of 82 Caucasian individuals, corresponding to 39 female individuals (48%) and 43 male individuals (52%). After analyzing the age distribution of all individuals, it was discovered that their ages ranged from 20 to 78 years, with an average age of 58 ± 12.8 years. The majority of individuals, *n* = 29 (35.4%), were found to be in the age group of 60 to 69 years old (Table 1).

Regarding the physical characteristics of the individuals, their average weight was 79.56 ± 25.8 kg, (45–175 kg). Their average height was 164.85 ± 9.50 cm (147–198 cm). The average BMI was 29.17 ± 8.49 kg/m^2^ (18.29–68.68 kg/m^2^). Among the BMI classes, overweight was the most common, accounting for 36.6% of the sample (*n* = 30). The individuals had an average abdominal circumference of 94.8 ± 13.3 cm (64–125 cm) (Table 1). 

### 3.1. Abdominal Circumference and Pericardial and Epicardial Fat

Of the 82 individuals, 45 belong to the Low PF group (54.9%). The remaining 37 individuals belong to the PF Alto group. On average, the Low PF group had a PF thickness of 3.1 ± 0.9 mm (1.4–4.5 mm) and a mean EF thickness of 3.6 ± 1.4 mm (1.5–7.1 mm). The High PF group had a mean PF thickness of 6.5 ± 1.8 mm (4.7–11.6 mm) and a mean EF thickness of 7.1 ± 2.6 mm, (2.1–13.1 mm) (Table 2).

Individuals belonging to the High PF group had a mean waist circumference of 101.07 cm, while those belonging to the Low PF group had a mean value of 89.62 cm (*p* < 0.0001) (Table 2).

### 3.2. Association between Abdominal Circumference and Age Group

An analysis was carried out to understand the relationship between waist circumference and age by sex. Figure 2 shows the distribution of average abdominal circumference by age group and sex. Men have a greater average waist circumference than women, increasing with age (*p* < 0.0001). There is a slight difference in the age group of 30 to 39 years, where females have a larger average waist circumference, but this is not statistically significant.

### 3.3. Assessment of Echocardiographic Variables and PF

As seen in Table 3, individuals in the High PF group had higher DT values (0.21 ms) than those in the Low PF group (0.18 ms), the difference being statistically significant (*p* = 0.013). The same trend was observed in the E/e’ ratio, where individuals with high levels of PF had a higher mean value (8.48) compared to those with low levels (6.40) (*p* < 0.0001).

The High PF group showed a decreased E/A ratio (0.94), while the Low FP group showed a higher E/A ratio of 1.10, indicating a statistically significant difference (*p* = 0.003).

The study found that individuals in the High PF group had lower mean values of septal e’ (6.64 cm/s) and lateral e’ (10.12 cm/s) compared to those in the Low PF group (9.58 cm/s) and (13.23 cm/s), the difference being statistically significant (*p* < 0.0001 and *p* < 0.0001).

Based on the observations carried out, it was found that individuals categorized as High PF had a higher VAE (average of 41.33 mL/m^2^), compared to the Low PF group, whose average was 32.8 mL/m^2^, the difference being statistically significant. Likewise, the TRJV was greater in the High PF group compared to the Low PF group (2.65 vs. 2.32 cm/s, *p* < 0.0001).

Statistically significant differences in LV diameter and mass were observed between the two groups (*p* = 0.014 and *p* = 0.003). The group with high PF levels had a mean LV diameter and LV mass of 54.89 mm and 183.57 g, respectively. The group with low PF levels had a mean LV diameter of 52.07 mm and LV mass of 149.64 g. 

In terms of LVEF, it was found that the group with high levels of PF had slightly lower mean values compared to the group with low levels of PF. This difference was not statistically significant (*p* = 0.093). A similar trend was observed for the echocardiographic parameter E, where the mean value was higher in the group with low PF levels compared to the group with high PF levels (*p* = 0.359).

### 3.4. Association between Abdominal Circumference and PF and EF Thickness

The Spearman correlation test was also used to better understand the association between abdominal circumference and PF and EF.

In Figure 3, we can see a clear correlation between the abdominal circumference and the amount of fat around the heart. This is shown through a scatter plot, where the abdominal circumference is the independent variable (x) and PF and EF are the dependent variables (y) in Figure 3A and 3B, respectively. The blue dots on the graph represent the measurements of the abdominal circumference and fat around the heart for each individual. By using the straight line, we can observe that as the abdominal circumference increases, both PF and EF also increase. These changes are statistically significant (*p* < 0.0001).

### 3.5. Association of Echocardiographic Variables with Pericardial Fat vs. Epicardial Fat

In Table 4, we present the characterization of the echocardiographic variables that were studied, including the E/A ratio, E, DT, TRJV, LV diameter, septal e’, lateral e’, E/e’ ratio, LAV, LVEF, and LV mass. These variables were analyzed concerning the PF and EF of all individuals. To better understand this association, the Spearman correlation test was used.

It has been observed that an increase in PF leads to a significant increase in the variables DT (*p* = 0.045), E/e’ (*p* < 0.0001), LAV (*p* < 0.0001), LV mass (*p* = 0.001), TRJV (*p* < 0.0001), and LV diameter (*p* = 0.041), indicating a positive correlation between PF and these variables. Conversely, a decrease in E/A (*p* = 0.001), e’ septal (*p* < 0.0001), and e’ lateral (*p* < 0.0001) has also been observed, indicating a negative correlation between these variables and PF (Table 4). An increase in EF showed a positive correlation with variables such as E/e’, LAV, LV mass, TRJV, and LV diameter. On the other hand, it showed a negative correlation with E/A, e’ septal, and e’ lateral.

### 3.6. Association between Pericardial Fat and Echocardiographic Variables, Multivariable Adjustment of Co-Factors

Considering the sensitivity of the diastolic parameter as well as the wide age range of the sample, a multivariable linear regression analysis was performed to examine the relationship between PF and the study’s echocardiographic variables, controlling the effects of sex, age, and BMI. After adjusting for these variables, all six parameters (septal e’, LAV, VRJ, LVEF, LV diameter, and LV mass) were statistically significant, as shown in Table 5.

### 3.7. Assessment between Pericardial Fat and the Diagnosis of Diastolic Function 

In Table 6, the relationship between the diagnosis of diastolic function and two groups of PF is demonstrated. To analyze this relationship in a better way, the Chi-square test was utilized. The table shows the number of individuals diagnosed with normal, undetermined diastolic function, and diastolic dysfunction. It also highlights the significance that exists between both groups.

## 4. Discussion

The measurement of PF is often overlooked during routine transthoracic echocardiography, as it is not a usual evaluation. However, PF thickness is a crucial predictor of numerous adverse cardiovascular outcomes and should not be ignored. It is important to note that it can indicate cardiovascular risk and should be carefully evaluated by clinicians. 

### 4.1. Sample Characterization

The participants in this study were evenly split between genders and across PF groups. The majority of participants were older, with the most common age range being 60–69. This is likely due to the increased need for healthcare in this age group. Additionally, most participants were overweight, which is a common characteristic of the population in a region of Central Portugal, called Beira Alta. These findings align with data from the 2018 Central Region of Portugal health profile, which showed a high prevalence of overweight individuals in the area [21].

### 4.2. Association of Abdominal Circumference with Age in Male and Female

The study also analyzed waist circumference, which was found to be greater in participants with higher levels of PF and EF. Results showed that abdominal circumference increased with age in both sexes, with males having higher index values. This is in line with previous research by Jennifer L Kuk et al. (2005), which suggests that metabolic changes that occur with age may explain this trend. The study also noted that hormonal differences between genders contribute to differences in abdominal circumference, which justifies the need for different cutoffs for males and females [22,23].

### 4.3. Relationship between Pericardial Fat and Abdominal Circumference

As per the literature and the results obtained, a significant increase in abdominal circumference was observed in individuals belonging to the PF High group. In a 2009 study, these findings were explained by the decrease in the production of adiponectin, which is a stabilizing hormone that inhibits NF-kB, in the presence of an increase in visceral fat. Due to this decrease, the activation of NF-kB increases, which leads to the production of TNF-a, causing more local inflammation and molecular aggregation. As a result, there is an increase in fat around the heart with an increase in visceral fat [24].

Several studies have found a positive correlation between abdominal circumference and PF and EF. This correlation confirms that waist circumference can be a predictive factor for increased fat around the heart [25,26].

### 4.4. Assessment of Echocardiographic Variables and Pericardial Fat

To investigate the relationship between PF and echocardiographic parameters, two groups were compared based on their PF and echocardiographic variables. The results showed that individuals with greater PF (PF High) had significantly lower mean values of septal e’, lateral e’, and the E/A ratio, while showing an increase in the mean values of E/e’ ratio, LAV, LV mass, TRJV, and LV diameter. These findings are consistent with a similar study by Vera H. W. de Wit Verheggen et al. (2020) that also separated their sample into two PF groups, and found that higher PF thickness influences some of the parameters of LV diastolic function, leading to a decrease in the mean values of septal e’, lateral e’, and the E/A ratio, as well as an increase in the E/e’ ratio, LAV, LV mass, and LV diameter [1,11].

The E/e’ ratio, septal e’, and lateral e’ are echocardiographic parameters that indicate the speed of left ventricular (LV) relaxation, which decreases due to some compressive condition affecting normal functioning, thus reducing its speed [11,27]. It is credible to assume that deregulation in the production of anti-inflammatory and anti-atherogenic cells, deficiency in the mediating enzyme of the renin–angiotensin–aldosterone system, and the migration of fibrosis-inducing cells are the mechanisms behind the development of atherosclerosis and consequent myocardial dysfunction. This suggests that the fat surrounding the heart could potentially trigger cardiovascular diseases. The compression of EF and PF on the myocardium has a significant impact on ventricular relaxation, leading to cardiac remodeling [28].

The E/A ratio and DT are two crucial parameters used to measure ventricular filling. When PF increases, both of these parameters tend to decrease and increase, respectively. Several studies by Vera H. W. de Wit Verheggen et al. (2020) [1] and Iacobellis et al. (2007) [29] reaffirm these results. According to Jin Seok Kim et al. (2021) [30], the E/A ratio decreases due to excess fat, which exerts a compressive mechanical force on the LV, affecting its normal relaxation. This leads to an increase in pressure within the LV at the start of the diastole, which affects the filling of the LV, causing slower passive filling (E wave) and faster active filling (A wave). This, in turn, increases the volume retained in the LA, leading to atrial dilatation. The increase in pressure also prolongs passive filling, increasing DT [30,31]. The infiltration of EF in the auricular wall not only predisposes the cavity to enlargement but also to the development of atrial fibrillation [32].

As a result of the increased difficulty that the right ventricle faces in overcoming the increase in pulmonary pressures during systole, the condition also affects the TRJV. As LV diastolic dysfunction worsens, the tricuspid reflux velocity increases progressively, making it a good marker of the hemodynamic impact of deficient left ventricular relaxation [33].

According to the study, individuals in the PF High group showed higher mean values of LV mass and LV diameter. This finding is consistent with another investigation published in 2021. The study explains that this relationship is due to the predisposition of these individuals to have greater blood volume because of the higher metabolic demand they face. This, in turn, leads to a greater cardiac output [34]. The increase in ventricular mass occurs as a result of the need to compensate for underlying hemodynamic changes, an increase in LV volume due to an increase in afterload, and as an adaptive mechanism in the face of changes in diastolic function. These changes are characteristic of individuals with high abdominal circumference [35].

The study also tested the relationship between systolic function and PF. Although LVEF showed lower mean values among individuals with higher PF, the relationship was not statistically significant between groups. This is unlike the findings of Gijs Van Woerden et al. (2018) [36], who concluded that there is a statistically significant correlation between PF and LVEF. The author explains that while there is a decrease in cardiac output as a result of deficient ventricular filling [36,37], fat produces high amounts of pro-inflammatory cytokines due to its contact with the cardiac surface. This results in damage to the myocardial cells. Additionally, fat promotes the abundance of free fatty acids, which, when integrated into the myocardial cell, promote toxic metabolism, leading to cell apoptosis (lipotoxicity) [38,39]. One possible explanation for the difference in results found in various studies could be that the values of PF thickness in the sample were lower than those found in other studies.

The relationship between fat and systolic ventricular function has been studied over time. Recently, a study using speckle tracking (transthoracic echocardiogram modality) tested the correlation between fat and systolic ventricular function in individuals with preserved left ventricular function. The results revealed a positive correlation between the two [40,41].

### 4.5. Association of Echocardiographic Variables with Pericardial Fat vs. Epicardial Fat

This study aimed to examine the relationship between PF and EF and echocardiographic variables. The analysis showed that PF had a strong correlation with nine echocardiographic variables, including DT, E/e’, LAV, LV Mass, TRJV, and LV Diam, which increased significantly, and E/A, septal e’, and lateral e’, which decreased significantly. Similarly, EF was found to have a statistically significant correlation with eight echocardiographic variables, including E/e’, LAV, LV Mass, TRJV, and LV Diameter, which increased significantly, and E/A, septal e’, and lateral e’, which decreased significantly. These findings suggest that PF has a greater impact on echocardiographic variables related to diastolic function, while EF has a greater influence on left ventricular systolic function.

The results of this study are consistent with the findings of Vera H. W. de Wit Verheggen et al. (2020) [1], who explained that PF exerts compressive mechanisms on ventricular relaxation due to its location. In contrast, Banafsheh Arshi et al. (2023) stated that EF, which is closer to the myocardium, affects systolic function due to the pro-inflammatory cytokines it releases in the myocardium, in addition to its effect on diastolic function. While LVEF was not statistically significant in this study, its values decreased substantially when correlated with EF, compared to when correlated with PF [41].

### 4.6. Association between Pericardial Fat and Echocardiographic Variables, Multivariable Adjustment of Co-Factors 

PF was found to be significantly associated with several parameters of diastolic function, including septal e’, LAV, VRJ, LVEF, LV diameter, and LV mass, even after adjustment for sex, age, and BMI. These associations suggest that PF has an impact on these parameters of diastolic function, despite the wide age range studied. As noted by Okura H. et al. (2009) [42], diastolic function may be sensitive to other comorbidities, such as age and sex. Vera HW de Wit Verheggen et al. (2020) [1] also found significant associations between PF and septal e’, lateral e’, VAE, TRJV, and E/e’ after adjusting for sex, age, and BMI. However, they found no significant correlations with LVEF, LV diameter, or LV mass. These variations in results highlight the differences that exist between studies [1,42].

### 4.7. Assessment between PF and the Diagnosis of Diastolic Dysfunction

The study found a correlation between PF and the diagnosis of diastolic function. The PF High group had a significantly higher percentage (45.9%) of individuals diagnosed with diastolic dysfunction compared to the PF Low group (6.7%). The majority of the PF Low group (93.3%) had a normal or undetermined diastolic function.

### 4.8. Limitations

The investigation was limited by insufficient information on participant history, risk factors, and medication use, as well as a lack of age differentiation, which could impact results.

## 5. Conclusions

Individuals with higher levels of PF tend to have a larger abdominal circumference, indicating poorer LV diastolic performance. Monitoring the abdominal circumference can thus help identify a higher concentration of fat around the heart (PF and EF), which in turn can be an early warning sign of possible changes in diastole. The good news is that measuring the abdominal circumference is a simple task for clinicians and transthoracic echocardiography is the most accurate way to assess the amount of fat surrounding the heart. By adopting this practice, healthcare providers can help prevent the development and progression of diastolic heart failure. This not only has a direct impact on the future quality of life of these individuals but also reduces the burden on healthcare systems. Based on the research findings, it can be concluded that PF has a greater impact on LV diastolic function compared to EF. However, it is important to consider both types of fat as a risk factor for heart failure. Although the study found some evidence regarding the LVEF, further research is required to explore the impact of EF on systolic function.

## Figures and Tables

**Figure 1 diagnostics-14-00702-f001:**
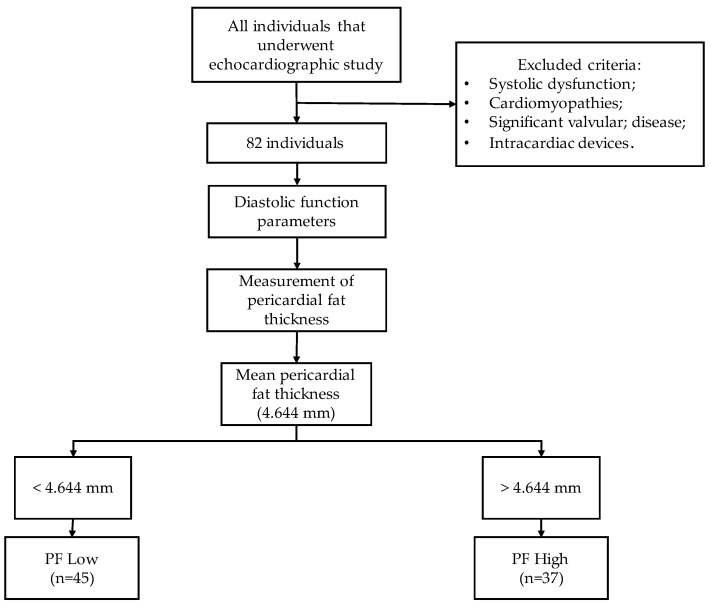
Sample organization.

**Figure 2 diagnostics-14-00702-f002:**
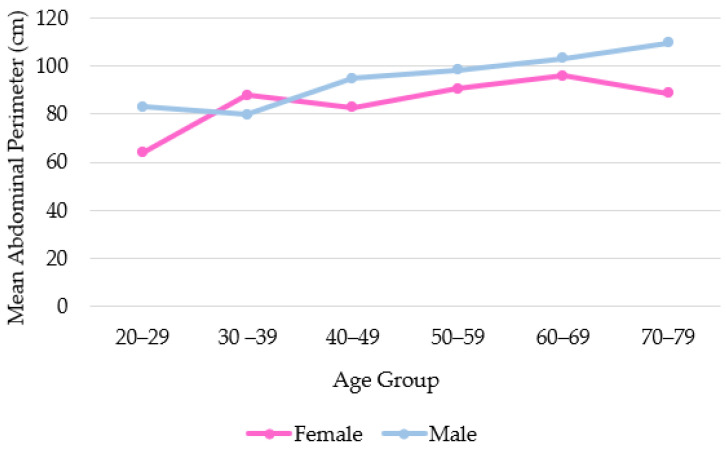
Distribution of abdominal perimeter by age.

**Figure 3 diagnostics-14-00702-f003:**
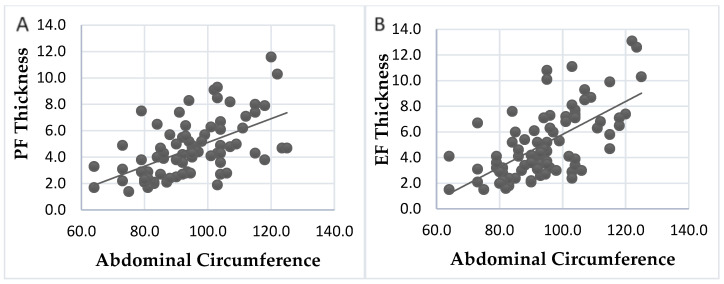
Association between abdominal perimeter and pericardial (**A**) and epicardial (**B**) Fat.

**Table 1 diagnostics-14-00702-t001:** Characteristics of the study participants.

Variables	Values
Sex	
Female (*n*, %)	39 (48%)
Male (*n*, %)	43 (52%)
Age (years)	58 ± 12.8
Age Group (*n*)	
20–29 y	2 (2.4%)
30–39 y	6 (7.3%)
40–49 y	13 (15.9%)
50–59 y	17 (20.7%)
60–69 y	29 (35.4%)
70–79 y	15 (18.3%)
Weight (kg)	79.56 ± 25.8
Height (cm)	164.85 ± 9.50
BMI (kg/m^2^)	29.17 ± 8.49
BMI Classes (*n*)	
Underweight	1 (1.2%)
Normal	25 (30.5%)
Overweight	30 (36.6%)
Obesity I	16 (19.5%)
Obesity II	5 (6.1%)
Obesity III	5 (6.1%)
Abdominal Circumference (cm)	94.8 ± 13.3

**Table 2 diagnostics-14-00702-t002:** PF and EF thickness and abdominal circumference.

	PF Low	PF High	*p*-Value
PF Thickness (mm)	3.1	6.5	<0.0001
EF Thickness (mm)	3.6	7.1	<0.0001
Abdominal Circumference (cm)	89.62	101.7	<0.0001
Total (*n*)	45	37	

Legend: Mean pericardial fat thickness (PF Thickness), Mean epicardial fat (EF Thickness).

**Table 3 diagnostics-14-00702-t003:** Relationship between echocardiographic variables and the two groups of PF.

	Total Population (*n* = 82)	PF Low (*n* = 45)	PF High (*n* = 37)	(*p*-Value)
E (cm/s) *	67.88 ± 16.57	69.41 ± 11.92	66.01 ± 20.92	*p* = 0.359
DT (ms) *	0.20 ± 0.05	0.18 ± 0.04	0.21 ± 0.063	*p* = 0.013
E/A **	1.03 ± 0.43	1.10 ± 0.34	0.94 ± 0.50	*p* = 0.003
septal e’ (cm/s) *	8.25 ± 2.45	9.58 ± 2.17	6.64 ± 1.70	*p* < 0.0001
lateral e’ (cm/s) *	11.82 ± 3.20	13.23 ± 3.30	10.12 ± 2.06	*p* < 0.0001
E/e’ *	7.34 ± 2.62	6.40 ± 2.01	8.48 ± 2.83	*p* < 0.0001
LAV (mL/m^2^) *	36.65 ± 8.33	32.80 ± 4.97	41.33 ± 9.22	*p* < 0.0001
LVEF (%) *	65.57 ± 5.62	66.52 ± 4.99	64.42 ± 6.18	*p* = 0.093
LV Mass (g) *	164.95 ± 52.34	149.64 ± 34.08	183.57 ± 63.99	*p* = 0.003
TRJV (cm/s) **	2.47 ± 0.29	2.32 ± 0.22	2.65 ± 0.26	*p* < 0.0001
LV Diameter (mm) *	53.34 ± 5.19	52.07 ± 4.03	54.89 ± 6.04	*p* = 0.014

Legend: * (*t*-test), ** (Mann–Whitney test), E/A ratio (E/A), E-curve velocity of transmitral flow (E), E-curve deceleration time (DT), relaxation velocity in the septum (e’ septal), relaxation velocity in the lateral wall (e’ lateral), E/e’ ratio (E/e’), left atrial volume (LAV), tricuspid regurgitation jet velocity (TRJV), left ventricular diameter (LV diameter), LV ejection fraction (LVEF), left ventricular mass (LV mass).

**Table 4 diagnostics-14-00702-t004:** Association between echocardiographic variables and pericardial and epicardial fat.

	Pericardial Fat			Epicardial Fat		
	Correlation	Correlation Degree	(*p*-Value)	Correlation	Correlation Degree	(*p*-Value)
E (cm/s)	−0.053	Non-Significant	*p* = 0.633	0.032	Non-Significant	*p* = 0.776
DT (ms)	0.222	Low	*p* = 0.045	0.208	Non-Significant	*p* = 0.061
E/A	−0.363	Moderate	*p* = 0.001	−0.403	Moderate	*p* < 0.0001
septal e’ (cm/s)	−0.686	High	*p* < 0.0001	−0.731	High	*p* < 0.0001
lateral e’ (cm/s)	−0.630	High	*p* < 0.0001	−0.662	High	*p* < 0.0001
E/e’	0.494	Moderate	*p* < 0.0001	0.592	High	*p* < 0.0001
LAV (mL/m^2^)	0.683	High	*p* < 0.0001	0.693	High	*p* < 0.0001
LVEF (%)	−0.141	Non-Significant	*p* = 0.207	−0.216	Non-Significant	*p* = 0.052
LV Mass (g)	0.359	Moderate	*p* = 0.001	0.397	Moderate	*p* < 0.0001
TRJV (cm/s)	0.629	High	*p* < 0.0001	0.610	High	*p* < 0.0001
LV Diam (mm)	0.227	Low	*p* = 0.041	0.282	Low	*p* = 0.010

Legend: E/A ratio (E/A), velocity of the transmitral flow E curve (E), E curve deceleration time (DT), relaxation velocity in the septum (e’ septal), relaxation velocity in the lateral wall (lateral e’), E/e’ ratio (E/e’), left atrial volume (LAV), tricuspid regurgitation jet velocity (TRJV), left ventricular diameter (LV diameter), LV ejection fraction (LVEF), left ventricular mass (LV mass).

**Table 5 diagnostics-14-00702-t005:** Multivariable linear regression between PF and echocardiographic variables.

Variables	Unadjusted Regression Coefficient (95% CI)	*p*-Value	Adjusted Regression Coefficient (95% CI) ª	*p*-Value
septal e’	−0.639	<0.0001	−0.189	0.031
LAV	0.613	<0.0001	0.457	<0.0001
TRJV	0.595	<0.0001	0.381	0.001
LVEF	−0.298	0.007	−0.330	0.024
LV Diameter	0.388	<0.0001	0.297	0.016
LV Mass	0.473	<0.0001	0.381	0.001

Legend: (ª—Adjusted for age sex and BMI).

**Table 6 diagnostics-14-00702-t006:** Relationship between the two groups of pericardial fat and diastolic function diagnosis.

	PF Low	PF High	*p*-Value
Normal	38	8	<0.0001
Undetermined	4	12	<0.0001
Diastolic Dysfunction	3	17	<0.0001

## Data Availability

The data resulted from consultation of clinical files and collection in accordance with the approval of the Ethics Committee. Also in accordance with the recommendations of the Ethics Committee, they cannot be made publicly available. However, they are archived, coded and will be made available to Researchers who show interest.

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
