# Peer review of "The Influence of Pericardial Fat on Left Ventricular Diastolic Function"

_diagnostics, 2024, doi:10.3390/diagnostics14070702_

Round 1

Reviewer 1 Report (Previous Reviewer 3)

Comments and Suggestions for Authors

In your responses, it would be better you explain more in detail so that the reviewer can easily find where and what was precisely corrected in the revised manuscript. It is very difficult where the corrections were made in response to the reviewer’s comments, as you simply responded such as “done” or “changed” in a single word.

It is very regretful that English grammar check is strongly recommended even in the revised version. 

Page 2, line 71. “in the increase of left atrium (LA)” – increase of the LA volume? LA size? Clarify this. In addition, you should “the” in front of the left atrium.

Figure 1. Do not use abbreviation in the Title of the figure. Explain abbreviations used in the figure separately. 

Page 4, lines 157-159. Too many “or”s in a sentence. Grammatical check is strongly recommended.

Table 1. Decimal points are still not unified.

Graph 1. What is the difference between figure and graphs in your manuscript? Unify them.

Graph 2. Give full names of any abbreviation you used in the figure (or graph). For example, “PF, pericardial fat; EF, epicardial fat.”

Tables 5 and 6. Provide full names of abbreviations you used in tables as foot notes.

Table 6. This table is poorly understandable as a sole. Title of the most left column is needed. 

Comments on the Quality of English Language

English grammar check is strongly recommended. 

Author Response

In your responses, it would be better you explain more in detail so that the reviewer can easily find where and what was precisely corrected in the revised manuscript. It is very difficult where the corrections were made in response to the reviewer’s comments, as you simply responded such as “done” or “changed” in a single word.

It is very regretful that English grammar check is strongly recommended even in the revised version. 

Answer: Thank you for sending your suggestions, which we have implemented. We revised English again, using technical English, since it is our second language and not our mother tongue.

Page 2, line 71. “in the increase of left atrium (LA)” – increase of the LA volume? LA size? Clarify this. In addition, you should “the” in front of the left atrium.

Answer: Corrected to “The authors propose that it may be due to a limitation in the left atrial (LA) size enlargement caused by excessive PF.”

Figure 1. Do not use abbreviation in the Title of the figure. Explain abbreviations used in the figure separately. 

Answer: Titles of all figures corrected.

Page 4, lines 157-159. Too many “or”s in a sentence. Grammatical check is strongly recommended.

Answer: Grammatical error corrected.

Table 1. Decimal points are still not unified.

Answer: All decimal points are now unified per figure.

Graph 1. What is the difference between figure and graphs in your manuscript? Unify them.

Answer: All graphs changed into figures.

Graph 2. Give full names of any abbreviation you used in the figure (or graph). For example, “PF, pericardial fat; EF, epicardial fat.”

Answer: All abbreviations were given full names.

Tables 5 and 6. Provide full names of abbreviations you used in tables as foot notes.

 Answer: Full names provided in foot notes.

Table 6. This table is poorly understandable as a sole. Title of the most left column is needed. 

Answer: Add the title in the left column and legend to make it more understandable.

Reviewer 2 Report (New Reviewer)

Comments and Suggestions for Authors

This revised manuscript aimed to determine whether pericardial and epicardial fat are associated with left ventricular diastolic function. I am not the reviewer for the original manuscript, but it looks like that the authors have carefully addressed many concerns pointed out from the reviewers for the original manuscript.

Author Response

Thank you very much for your comments.

Reviewer 3 Report (New Reviewer)

Comments and Suggestions for Authors

The work makes a very favorable impression, the design of the study is well thought out, the research methods necessary for this analysis are applied, taking into account the type of distribution of the analyzed features (their normality is estimated) and the necessary covariates (gender, age, BMI), the results of the study are consistently and reasonably described, well illustrated with tables and figures. Thus, the conclusion made by the authors in the work (Measurement of abdominal circumference, PF and EF is an early indicator of diastolic changes, and transthoracic echocardiography is the gold standard of examination) has been convincingly proven and is beyond doubt.

There are two questions about work

1. Did the authors determine the required sample size for this study and, if so, how did they do it?

2. Have the authors analyzes the signs they studied in groups depending on BMI (normal, overweight, obesity)?

In general, the work with minor clarifications can be recommended for publication.

Author Response

The work makes a very favorable impression, the design of the study is well thought out, the research methods necessary for this analysis are applied, taking into account the type of distribution of the analyzed features (their normality is estimated) and the necessary covariates (gender, age, BMI), the results of the study are consistently and reasonably described, well illustrated with tables and figures. Thus, the conclusion made by the authors in the work (Measurement of abdominal circumference, PF and EF is an early indicator of diastolic changes, and transthoracic echocardiography is the gold standard of examination) has been convincingly proven and is beyond doubt.

There are two questions about work

  1. Did the authors determine the required sample size for this study and, if so, how did they do it?

Answer: The sample was based on all individuals who, during the study period, underwent the examination at the aforementioned hospital.

  1. Have the authors analyzes the signs they studied in groups depending on BMI (normal, overweight, obesity)?

Answer: We considered the fact that pericardial and epicardial fat are positively correlated with abdominal circumference, height, weight, and BMI. As a result, there was a clear difference between our two groups in terms of their BMI. The PF Low group had a lower BMI than the PF High group. Although it would have been an excellent analysis, we did not examine our echocardiography variables for each BMI class. Instead, we focused on abdominal circumference as it is an important and straightforward measurement for clinicians to be aware of.

Round 2

Reviewer 1 Report (Previous Reviewer 3)

Comments and Suggestions for Authors The manuscript has been improved via revision.    One minor comment has remained: Table 6, Declare what the values in the table present. Is it the number of patients?    Once the above comment is properly answered, the revised manuscript may be acceptable for publication.  Comments on the Quality of English Language The manuscript has been improved via revision.    One minor comment has remained: Table 6, Declare what the values in the table present. Is it the number of patients?    Once the above comment is properly answered, the revised manuscript may be acceptable for publication. 

Author Response

This table displays the diagnosis of diastolic function and the number of patients with each diagnosis. We add "n" in both the table and the legend.

This manuscript is a resubmission of an earlier submission. The following is a list of the peer review reports and author responses from that submission.

Round 1

Reviewer 1 Report

Comments and Suggestions for Authors

In the manuscript, 82 patients had their abdominal circumference measured and underwent transthoracic echocardiography to measure the thickness of pericardial and epicardial fat and assess the left ventricular diastolic function. In the results, authors thought that measurement of abdominal circumference, pericardial and epicardial fat is an early indicator of diastolic changes with transthoracic echocardiography being the gold standard exam. It is an interesting but primeval study. Some individual comments were provided below:

Major

1)In the manuscript, the information from graphs were very limited. Put the information into a table is more appropriate.

2)The study need add a validation population to verify the results. The sample size is small.

Minor

In table 1, p value=0,0,13 of DT is a clerical error. 

Comments on the Quality of English Language

Minor editing of English language required.

Author Response

Review 1

Major

1)In the manuscript, the information from graphs were very limited. Put the information into a table is more appropriate.

Answer: All graphs were turned into tables, only graphs 5 and 7 remain and were changed to graph1 and 2, respectively.

2)The study needs add a validation population to verify the results. The sample size is small.

The Seia Hospital, where the sample was collected, serves a population of less than twenty thousand people. Considering the pathology and all the surroundings, it seems to us that it is appropriate.

Reviewer 2 Report

Comments and Suggestions for Authors

The Influence of Pericardial Fat on Left Ventricular Diastolic Function.

The manuscript entitled “The Influence of Pericardial Fat on Left Ventricular Diastolic Function.” by Duarte et al., is an interesting article. Here, the authors measured measure the thickness of pericardial and epicardial fat and assess the left ventricular diastolic function and demonstrated that the measurement of abdominal circumference, pericardial and epicardial fat is an early indicator of diastolic changes with transthoracic echocardiography being the gold standard exam.

Overall, the information presented in this manuscript is useful and I approve its publication after some major updates. 

Minor comments: I suggest that these comments be updated before publication.

a. In the methods section, it is essential to incorporate details about ethical clearance and the year of sample collection. Inclusion criteria and exclusion criteria should also be clearly outlined. Additionally, the current sample size is insufficient; therefore, it is recommended to increase the sample size.

b. The experimentation aspect needs to be expanded. Consider incorporating biochemical analyses such as testing serum creatinine and cystatin C levels, as well as assessing total cholesterol, triglycerides, and other relevant parameters.

c. Ensure that figures are formatted using GraphPad Prism to meet publication standards.

d. The rationale and significance of the study require improvement. Please revise and enhance these aspects for a more comprehensive understanding.

Comments on the Quality of English Language

Need to be improved

Author Response

Review 2

Minor comments

  1. a) In the methods section, it is essential to incorporate details about ethical clearance and the year of sample collection. Inclusion criteria and exclusion criteria should also be clearly outlined. Additionally, the current sample size is insufficient; therefore, it is recommended to increase the sample size.

Answer: Details about the clearance and the year of sample collection, incorporated in methods, “study design” section and “ethics” section. Inclusion criteria and exclusion criteria are incorporated in Fig. 1.

The Seia Hospital, where the sample was collected, serves a population of less than twenty thousand people. Considering the pathology and all the surroundings, it seems to us that it is appropriate.

  1. The experimentation aspect needs to be expanded. Consider incorporating biochemical analyses such as testing serum creatinine and cystatin C levels, as well as assessing total cholesterol, triglycerides, and other relevant parameters.

Answer: At this stage it is not possible to access data that is not covered by the Ethics Commit.

  1. c) Ensure that figures are formatted using GraphPad Prism to meet publication standards.

Answer: All tables and graphs were formatted to meet publication standards. Some graphs became tables to better understanding.

  1. The rationale and significance of the study require improvement. Please revise and enhance these aspects for a more comprehensive understanding.

Answer: The rationale and significance of the study were improved in introduction and in discussion.

Reviewer 3 Report

Comments and Suggestions for Authors

Re: The influence of Pericardial Fat on Left Ventricular Diastolic Function

 Is there any difference between pericardial fat and epicardial fat in your manuscript? If there is, clarify and define the two terms, or if there is not, unify the duplicated terms throughout the manuscript. 

 There is an important issue in statistical analysis. To assess the relationship between echocardiographic variables (diagnosis of diastolic dysfunction) and pericardial fat, multivariable adjustment of co-factors of diastolic dysfunction are required. 

Abstract: 

 In introduction (page 3, lines 97-99), you described that you aimed to establish a correlation between abdominal circumference and pericardial/epicardial fat and the latter with LV diastolic performance. However, the results of the abstract do not contain these correlations. 

 Page 1, line 21. Designate the abbreviation (PF) when you use it in the first time. 

Introduction:

 Page 2, lines 55-56. “Several studies ..”, provide adequate citation for this sentence. 

 Page 2, lines 57-82. These two paragraphs are too long. You may simplify the main concept to show in Introduction section, and then move the details to Methods or Discussion sections. 

 Page 2, lines 85-86. Describe the abbreviations of EFT1 and EFT2. 

Materials and Methods:

 2.1. Study Design. Declare the study period in which study participants were actually enrolled in the study. 

 Page 3, line 109. “all individuals”. What type of individuals were included in the study? Were they cardiovascular patients with heart disease or healthy subjects who underwent echocardiography for health check-up?

 Page 4, lines 143-145. You divided the subjects into two groups based on the mean value of pericardial fat thickness. What was the data distribution of measurements of pericardial fat thickness? What was the range of the measurements?

 Page 4, lines 143-145. Provide the rationale of using the mean value to dichotomize patient group.

 Page 4, lines 143-145. Define abbreviation before its first use (PF). 

 2.4. Statistical. Use “Statistics” or “Statistical analysis” rather than using adjective form in subtitle.

 Page 4, line 156. “T-Student test”, change to “Student’s t-test”.

 Page 5. 2.5. “Ethic”, Change to “Ethics”.

Results:

 Graph 2. Indicate names of x and y axis.

 Page 6, lines 191-196. Provide range or IQR in parenthesis, rather than presenting the highest and the smallest values, respectively.

 Page 6, line 194. Can a BMI be 68.68 kg/m2?

 Graphs 1-3. Data shown in graphs 1-3 do not deserve to be depicted in graphs, as they are simple baseline patient characteristics. They would rather be presented in tables. 

 Graph 3. Is this title of graph correct?

 Page 6, lines 202-208. Indicate the range of data in parenthesis, rather than describing the maximum and minimum values, respectively. 

 Page 6, lines 208-210. Is this sentence for all cohort? Please clarify. 

 Graph 4. Check the spelling of x axis.

 Page 7, lines 214-216. Was there any statistical or clinically meaningful difference in abdominal circumference according to the age groups and sex? Results section should contain numerical data and meaningful statistics, not just showing analysis you conducted without output. 

 Page 7, lines 220-223. Simplify the description. The p value can be presented in parenthesis at the end of a sentence.

 Graph 6. This graph is unnecessary as it is repetitive of description in the manuscript. 

 Page 8, lines 227-239. Do not just repeat the data which is already shown in the table.

 Page 8, lines 242-246. This sentence can be rewrote in more sophisticated form as follows: 
Similarly, TRJV was higher in the PF High group compared to the PF Low group (2.65 vs. 2.32 cm/s, p < 0.0001).

 Page 8, lines 247-251. In similar with the above mention, you can also simplify and rewrite this paragraph more academically. 

 Table 1. Unify the decimal places in each parameter.

 Table 1, Legend. T-Student test, change to Student’s t-test.

 Graph 7. The legend below the title of Graph 7 looks unnecessary.

 Page 10, lines 286-290. If there was a positive or negative correlation, you should provide the statistical results from correlation analysis with numerical and scientific data and p values. OR if they are already presented in Table, you should indicate which table contains the details using parenthesis. 

 Page 10, lines 294-295. You may delete this sentence as this is redundant. You can just indicate Table 2 using parenthesis following the relevant description above. 

 Table 2. Use English. 

 Page 11, line 303. Do not use abbreviation in the subtitle (DD).

Discussion:

 Page 12, lines 314-135. Give sufficient evident for this sentence. 

 Page 12, lines 315-322. This part is repetitive with Introduction.

 Page 12, line 328. Which region? Specify the region you want to comment on.

 Page 13, lines 349-359. To assess the relationship between echocardiographic variables (diagnosis of diastolic dysfunction) and pericardial fat, multivariable adjustment of co-factors of diastolic dysfunction, such as age, physical activity, smoking, BMI, diabetes, hypertension, and coronary artery disease, are required. Please refer to your reference #21 (Mahabadi et al, 2009).

Comments on the Quality of English Language

Use English throughout the manuscript including tables. 

Author Response

Review 3

 Is there any difference between pericardial fat and epicardial fat in your manuscript? If there is, clarify and define the two terms, or if there is not, unify the duplicated terms throughout the manuscript. 

Answer: Yes, there is a difference, and it is now clarified in introduction page 2 line 50.

There is an important issue in statistical analysis. To assess the relationship between echocardiographic variables (diagnosis of diastolic dysfunction) and pericardial fat, multivariable adjustment of co-factors of diastolic dysfunction are required. 

Answer: To overcome this issue, we now, did a multivariable adjustment for sex, age and BMI. It is explained in methods page 4 line 157-160 and included in results 3.6 and in discussion 4.6.

Abstract

In introduction (page 3, lines 97-99), you described that you aimed to establish a correlation between abdominal circumference and pericardial/epicardial fat and the latter with LV diastolic performance. However, the results of the abstract do not contain these correlations. 

Answer: The abstract now contain the correlations performed in the study.

Introduction:

Page 2, lines 55-56. “Several studies ..”, provide adequate citation for this sentence. 

Answer: References provided.

Page 2, lines 57-82. These two paragraphs are too long. You may simplify the main concept to show in Introduction section, and then move the details to Methods or Discussion sections. 

Answer: Paragraphs simplify and details moved to methods.

Page 2, lines 85-86. Describe the abbreviations of EFT1 and EFT2

Answer: Abbreviations explained.

Materials and Methods

2.1. Study Design. Declare the study period in which study participants were actually enrolled in the study. 

Answer: Period add to study design, page 3 line 104.

Page 3, line 109. “all individuals”. What type of individuals were included in the study? Were they cardiovascular patients with heart disease or healthy subjects who underwent echocardiography for health check-up? Healthy

Answer: All individuals were healthy who underwent an echocardiography for a health check-up. Add to the study design that all individuals were healthy page 3 line 101.

Page 4, lines 143-145. You divided the subjects into two groups based on the mean value of pericardial fat thickness. What was the data distribution of measurements of pericardial fat thickness? What was the range of the measurements?

Answer: Range of measurements add in methods 2.2 protocol, page 4 line 140.

Page 4, lines 143-145. Provide the rationale of using the mean value to dichotomize patient group.

 Answer: Rationale of using the mean value described in protocol page 4 line 136-140.

Page 4, lines 143-145. Define abbreviation before its first use (PF). 

Answer: All abbreviations clarified.

2.4. Statistical. Use “Statistics” or “Statistical analysis” rather than using adjective form in subtitle.

Answer: Corrected to Statistical analysis.

Page 4, line 156. “T-Student test”, change to “Student’s t-test”.

Answer: Corrected.

Page 5. 2.5. “Ethic”, Change to “Ethics”.

Answer: Corrected.

Results:

Graph 2. Indicate names of x and y axis.

Answer: Graph 2 turned into table.

Page 6, lines 191-196. Provide range or IQR in parenthesis, rather than presenting the highest and the smallest values, respectively.

Answer: All ranges provided.

Page 6, line 194. Can a BMI be 68.68 kg/m2?

Answer: Yes, the subject in question has a weight of 165 kg and a height of 155 cm.

Graphs 1-3. Data shown in graphs 1-3 do not deserve to be depicted in graphs, as they are simple baseline patient characteristics. They would rather be presented in tables. 

Answer: Graphs 1 to 3 turned into tables.

Graph 3. Is this title of graph correct?

Answer: It was not, but now is changed to table.

Page 6, lines 202-208. Indicate the range of data in parenthesis, rather than describing the maximum and minimum values, respectively. 

Answer: Range of data indicated in parenthesis.

Page 6, lines 208-210. Is this sentence for all cohort? Please clarify. 

Answer:  Yes it was, but is now moved to page 5 line 188 for better understanding.

Graph 4. Check the spelling of x axis.

Answer: Turned into table.

Page 7, lines 214-216. Was there any statistical or clinically meaningful difference in abdominal circumference according to the age groups and sex? Results section should contain numerical data and meaningful statistics, not just showing analysis you conducted without output. 

Answer: Yes, there was. This paragraph now contains numerical data and meaningful statistics.

Page 7, lines 220-223. Simplify the description. The p value can be presented in parenthesis at the end of a sentence.

 Answer: Simplified description.

Graph 6. This graph is unnecessary as it is repetitive of description in the manuscript.

Answer: Graph 6 erased.

Page 8, lines 227-239. Do not just repeat the data which is already shown in the table.

Answer: Simplified description.

Page 8, lines 242-246. This sentence can be rewrote in more sophisticated form as follows: 
Similarly, TRJV was higher in the PF High group compared to the PF Low group (2.65 vs. 2.32 cm/s, p < 0.0001).

Answer: The sentence rewrote as suggested.

Page 8, lines 247-251. In similar with the above mention, you can also simplify and rewrite this paragraph more academically. 

Answer: The paragraph rewrote in a simpler and academically way.

Table 1. Unify the decimal places in each parameter.

Answer: Done.

Table 1, Legend. T-Student test, change to Student’s t-test.

Answer: Changed.

Graph 7. The legend below the title of Graph 7 looks unnecessary.

Answer: Legend erased, now graph 2.

Page 10, lines 286-290. If there was a positive or negative correlation, you should provide the statistical results from correlation analysis with numerical and scientific data and p values. OR if they are already presented in Table, you should indicate which table contains the details using parenthesis. 

Answer: Add a reference to the table in question, as it was already presented in table.

Page 10, lines 294-295. You may delete this sentence as this is redundant. You can just indicate Table 2 using parenthesis following the relevant description above. 

Answer: Sentence erased, as suggested.

Table 2. Use English. 

Answer: Corrected.

Page 11, line 303. Do not use abbreviation in the subtitle (DD).

Answer: Corrected

Discussion:

 Page 12, lines 314-135. Give sufficient evident for this sentence. 

Answer: Paragraph rewrote, and new references add for evidence.

Page 12, lines 315-322. This part is repetitive with Introduction.

Answer: Erased and paragraph rewrote.

Page 12, line 328. Which region? Specify the region you want to comment on.

Answer: Central Region of Portugal. Add to the paragraph.

Page 13, lines 349-359. To assess the relationship between echocardiographic variables (diagnosis of diastolic dysfunction) and pericardial fat, multivariable adjustment of co-factors of diastolic dysfunction, such as age, physical activity, smoking, BMI, diabetes, hypertension, and coronary artery disease, are required. Please refer to your reference #21 (Mahabadi et al, 2009).

Answer: To overcome this issue, we now, did a multivariable adjustment for sex, age and BMI. It is explained in methods page 4 line 157-160 and included in results 3.6 and in discussion 4.6.